# Topologically enhanced harmonic generation in a nonlinear transmission line metamaterial

You Wang[1], Li-Jun Lang [1,2], Ching Hua Lee[3,4], Baile Zhang [1,5] & Y.D. Chong [1,5]

Nonlinear transmission lines (NLTLs) are nonlinear electronic circuits used for parametric amplification and pulse generation, and it is known that left-handed NLTLs support enhanced harmonic generation while suppressing shock wave formation. We show experimentally that in a left-handed NLTL analogue of the Su-Schrieffer-Heeger (SSH) lattice, harmonic generation is greatly increased by the presence of a topological edge state. Previous studies of nonlinear SSH circuits focused on solitonic behaviours at the fundamental harmonic. Here, we show that a topological edge mode at the first harmonic can produce strong propagating higher-harmonic signals, acting as a nonlocal cross-phase nonlinearity. We find maximum third-harmonic signal intensities five times that of a comparable conventional left-handed NLTL, and a 250-fold intensity contrast between topologically nontrivial and trivial configurations. This work advances the fundamental understanding of nonlinear topological states, and may have applications for compact electronic frequency generators.

[1] Division of Physics and Applied Physics, School of Physical and Mathematical Sciences, Nanyang Technological University, Singapore 637371, Singapore. [2] Guangdong Provincial Key Laboratory of Quantum Engineering and Quantum Materials, SPTE, South China Normal University, Guangzhou 510006, China. [3] Institute of High Performance Computing, A*STAR, Singapore 138632, Singapore. [4] Department of Physics, National University of Singapore, Singapore 117551, Singapore. [5] Centre for Disruptive Photonic Technologies, Nanyang Technological University, Singapore 637371, Singapore. Correspondence and requests for materials should be addressed to B.Z. (email: blzhang@ntu.edu.sg) or to Y.D.C. (email: yidong@ntu.edu.sg)

Topological edge states—robust bound states guaranteed to exist at the boundary between media with topologically incompatible band structures—were first discovered in condensed matter physics[1]. Recently, electronic *LC* circuits have emerged as a highly promising method of realizing these remarkable phenomena[2–9]. Compared to other classical platforms like photonics[10–13], acoustics[14–16], and mechanical lattices[17–19], which have also been used to realize topologically nontrivial band structures and topological edge states, electronic circuits have several compelling advantages: extreme ease of experimental analysis; the ability to fabricate complicated structures via printed circuit board (PCB) technology; and the intriguing prospect of introducing nonlinear and/or amplifying circuit elements to easily study how topological edge states behave in novel physical regimes. Notably, circuits have been used to study the Su–Schrieffer–Heeger (SSH) chain (the simplest one-dimensional topologically-nontrivial lattice)[4,20], nonlinear SSH chains supporting solitonic edge states[8], two-dimensional topological insulator lattices[2], and the corner states of high-order topological insulators[5,9].

One of the most interesting questions raised by the emergence of topologically nontrivial classical lattices is how topological edge states interact with nonlinear media. Previous studies have focused on nonlinearity-induced local self-interactions in the fundamental harmonic, which can give rise to solitons with anomalous plateau-like decay profiles in nonlinear SSH chains[8,21], or chiral solitons in two-dimensional lattices[22–27]. It has also been suggested that topological edge states in nonlinear lattices could be used for robust traveling-wave parametric amplification[28], optical isolation[29], and other applications[30–33].

In this paper, we report on the implementation of a nonlinear SSH chain based on a left-handed nonlinear transmission line (NLTL)[34–41], in which the topological edge state induces highly efficient harmonic generation. Although previous studies have emphasized the role of local self-interactions, including in a previous demonstration of a nonlinear SSH circuit based on weakly-coupled *LC* resonators[8], an important feature of our circuit is the decisive role of higher-harmonic signals in modulating the first-harmonic modes: they can drive the entire lattice, not just the edge, deeper into the nontrivial regime at the first-harmonic frequencies. This behavior is aided by the fact that the left-handed NLTL has an unbounded dispersion curve supporting traveling-wave higher-harmonic modes[38–42].

Our measurements on the nonlinear circuit reveal a first-harmonic mode that is localized to the lattice edge, similar to a linear topological edge state, as well as higher-harmonic waves that propagate into the lattice bulk and have voltage amplitudes reaching over an order of magnitude larger than the first-harmonic signal. The intensity of the generated third-harmonic signal has a maximum of ≈2.5 times that of the input first-harmonic signal, compared to <0.5 for a comparable conventional left-handed NLTL without a topological edge state. The important role played by the topological edge state is further demonstrated by the fact that the third-harmonic intensity is 250 times larger than in a trivial circuit, which has equivalent parameters but lacks a topological edge state in the linear limit, using the same input parameters.

## Results

**Circuit design.** The transmission line circuit is shown schematically in Fig. 1a. It contains inductors of inductance $L$ and capacitors of alternating (dimerized) capacitances $C_a$ and $C_b$. We will shortly treat the case where the $C_b$ capacitors are nonlinear (the $L$ and $C_a$ elements are always linear). First, consider the linear limit where $C_b$ is a constant. We define the characteristic angular frequency $\omega_a = (LC_a)^{-1/2}$, and the capacitance ratio

$$\alpha = C_a/C_b. \qquad (1)$$

The case of $\alpha = 1$ corresponds to a standard (non-dimerized) left-handed transmission line. This type of transmission line is characterized by having sites separated by capacitors, and connected to ground by inductors, rather than vice versa. Left-handed NLTLs have been shown to be useful for parametric amplification and pulse generation[38–41].

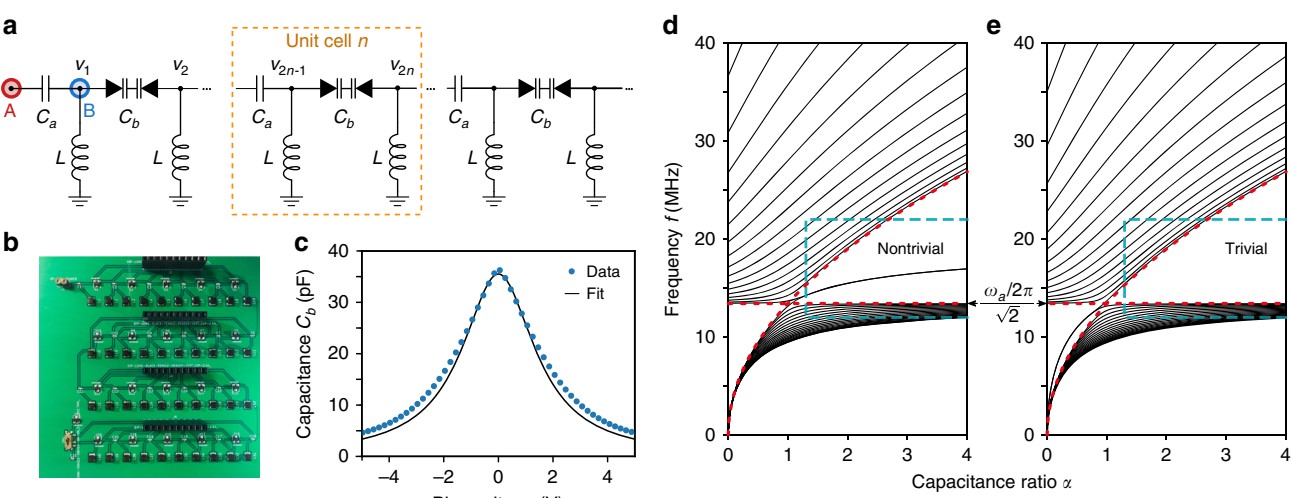

**Fig. 1** Design and implementation of SSH-like left-handed transmission line. **a** Schematic of the transmission line circuit with alternating capacitances: linear capacitors with capacitance $C_a$, and back-to-back varactors with nonlinear capacitance $C_b$. The capacitances act like hoppings in a nonlinear Su–Schrieffer–Heeger (SSH) model. An input voltage is applied at points A or B to probe the topologically trivial or nontrivial lattice. **b** Photograph of the printed circuit board. **c** Capacitance $C_b$ versus bias voltage. Dots are calculated from varactor manufacturer data, and the solid curve is the fit based on Eq. (5). **d**, **e** Calculated eigenfrequencies of a finite closed linear circuit, versus the capacitance ratio $\alpha = C_a/C_b$. The characteristic frequency is $f_a = \omega_a/2\pi \approx 19$ MHz, and the lattice has 40 sites. Two cases are shown: **d** $C_a$-type capacitors at the edge, for which the $\alpha > 1$ gap is topologically nontrivial; **e** $C_b$-type capacitors at the edge, for which the $\alpha > 1$ gap is trivial. Red dotted curves indicate the band-edge frequencies $f_a/\sqrt{2}$ and $\sqrt{\alpha/2}f_a$. Blue dashes indicate the operating regime of the nonlinear circuit, with $\alpha \approx 1.3$ in the linear (low-voltage) limit and $\alpha$ effectively increasing with voltage amplitude

Let us treat the points adjacent to the capacitors as lattice sites, indexed by an integer $k$, and close the circuit by grounding the edges [the left edge is the site labeled A in Fig. 1a]. Using Kirchhoff's laws, we can show that a mode with angular frequency $\omega$ satisfies (see Supplementary Note 1):

$$\left(\mathcal{H} - \frac{1}{\alpha}\right) \begin{pmatrix} v_1 \\ v_2 \\ v_3 \\ \vdots \end{pmatrix} = \left(1 - \frac{\omega_a^2}{\omega^2}\right) \begin{pmatrix} v_1 \\ v_2 \\ v_3 \\ \vdots \end{pmatrix}, \quad (2)$$

where $v_k$ denotes the complex voltage on site $k$. The matrix $\mathcal{H}$ has the form of the SSH Hamiltonian:

$$\mathcal{H} = \begin{pmatrix} 0 & \frac{1}{\alpha} & & & \\ \frac{1}{\alpha} & 0 & 1 & & \\ & 1 & 0 & \frac{1}{\alpha} & \\ & & \frac{1}{\alpha} & 0 & \ddots \\ & & & \ddots & \ddots \end{pmatrix}. \quad (3)$$

Thus, the eigenfrequency modes of the circuit have a one-to-one correspondence with the SSH eigenstates.

The band diagram for the linear closed circuit is shown in Fig. 1d. The lack of an upper cutoff frequency is a characteristic of left-handed transmission lines[40]. There is a bandgap in the range $\omega_a/\sqrt{2} < \omega < \sqrt{\alpha/2}\,\omega_a$. For $\alpha > 1$, the bandgap contains edge states, which are zero-eigenvalue eigenstates of $\mathcal{H}$ that can be characterized via a topological invariant derived from the Zak phase[1]. The edge state's angular frequency is

$$\omega_{es} = \sqrt{\alpha/(1+\alpha)}\,\omega_a. \quad (4)$$

Note that the edge state are not at zero frequency, nor do they lie at precisely the middle of the bandgap; this is due to the aforementioned mapping from the circuit equations to the SSH model—specifically, the fact that $\omega$ is not the eigenvalue in Eq. (2).

For $\alpha < 1$, there is a finite bandgap below $\omega_a/\sqrt{2}$, which is topologically trivial and contains no edge states. If we swap the two types of capacitors, so that the $C_b$-type capacitors are the ones at the edge, then the $\alpha > 1$ bandgap is trivial and the $\alpha < 1$ bandgap nontrivial, as shown in Fig. 1e.

Next, consider a nonlinear circuit with each $C_b$ capacitor consisting of a pair of back-to-back varactors. The nonlinear capacitance $C_b$ decreases with the magnitude of the bias voltage (the voltage between the end-points of the capacitor), as shown in Fig. 1c. For theoretical analyses, it is convenient to model this nonlinearity by

$$\alpha_{\mathrm{nl}}(t) \approx A + B[\Delta V(t)]^2, \quad (5)$$

where $\alpha_{\mathrm{nl}}(t) \equiv C_a/C_b(t)$, and $\Delta V(t)$ is the bias voltage. The key feature of the nonlinearity is that at higher voltages, the effective value of $\alpha$ increases. Depending on the chosen boundary conditions, this drives the circuit deeper into the topologically trivial or nontrivial regime.

**Experimental results**. The implemented NLTL, shown Fig. 1b, contains a total of 40 sites, or 20 unit cells. The linear circuit elements have $L = 1.5\,\mu\mathrm{H}$ and $C_a = 47\,\mathrm{pF}$, so that $\omega_a/2\pi \approx 19\,\mathrm{MHz}$. By fitting Eq. (5) to manufacturer data for the varactors at low bias voltages (see Supplementary Note 3), we obtain

$A = 1.32$ and $B = 0.51\,\mathrm{V}^{-2}$ (thus, in the linear limit, $\alpha \approx 1.3 > 1$). The fitted capacitance–voltage relation is shown in Fig. 1c.

We supply a continuous-wave sinusoidal input voltage signal, with tunable frequency $f_{\mathrm{in}}$ and amplitude $V_{\mathrm{in}}$, to either of the points labeled A and B in Fig. 1a. This allows us to study the cases corresponding to Fig. 1d, e, which we refer to as the "nontrivial" and "trivial" lattices, respectively (see Methods). In both cases, the input site is denoted as $k = 0$.

A typical set of measurement results is shown in Fig. 2a–c, for $f_{\mathrm{in}} = 16\,\mathrm{MHz}$ and $V_{\mathrm{in}} = 2.5\,\mathrm{V}$. On each site $k$, the spectrum of the voltage signal is shown in Fig. 2c, with prominent peaks at odd harmonics ($f_{\mathrm{in}}$, $3f_{\mathrm{in}}$, $5f_{\mathrm{in}}$, etc.); even harmonics are suppressed due to the symmetry of the capacitance–voltage relation[39]. Focusing on the first and third harmonics, we define the respective peak values as $\left|v_k^f\right|$ and $\left|v_k^{3f}\right|$, and use these to plot Fig. 2a, b. We verified that these experimental data agree well with results from the SPICE circuit simulator (see Supplementary Note 5).

From Fig. 2a, b, we see that the nontrivial and trivial lattices exhibit very different behaviors for both the first- and third-harmonic signals. First, consider the first-harmonic signal. In both lattices, there is an exponential decay away from the edge, but the decay is sharper in the nontrivial lattice, which may be attributed to the enhanced intensity arising from the coupling of the input signal to the topological edge state. As a quantitative measure of the localization of the first-harmonic signal, Fig. 2d, e shows the inverse participation ratio (IPR) $\sum_k \left|v_k^f\right|^4 / \left(\sum_k \left|v_k^f\right|^2\right)^2$; a larger IPR corresponds to a more localized profile[43]. We see that the IPR is substantially larger in the nontrivial lattice than in the trivial lattice, over a broad range of $f_{\mathrm{in}}$ and $V_{\mathrm{in}}$. The strong difference in localization is a key signature of nonlinearity: in the linear regime, a driving voltage on the edges of the nontrivial and trivial lattices would produce different overall amplitudes, but the same exponential decay profile (see Supplementary Note 1). It is interesting to note that the region of enhanced IPR, shown in Fig. 2d, closely resembles the nontrivial bandgap in Fig. 1d.

We can also see from Fig. 2a, c that strong higher-harmonic signals are present in the nontrivial lattice. Moreover, Fig. 2a indicates that the third-harmonic signal is extended, not localized to the edge. To understand this in more detail, we define

$$\chi = \left\langle \left|v_k^{3f}\right|^2 \right\rangle / V_{\mathrm{in}}^2, \quad (6)$$

which quantifies the intensity of the third-harmonic signal relative to the input intensity at the first harmonic. Here, $\langle \cdots \rangle$ denotes an average over the first ten lattice sites. Figure 3a, b plots the variation of $\chi$ with $f_{\mathrm{in}}$ and $V_{\mathrm{in}}$. In the nontrivial circuit, the maximum value of the normalized intensity is $\chi \approx 2.5$ for $f_{\mathrm{in}} \sim 16$ MHz and $1\,\mathrm{V} \lesssim V_{\mathrm{in}} \lesssim 4\,\mathrm{V}$. The fact that $\chi$ peaks over a relatively narrow frequency range, as shown in Fig. 3a, may be a finite-size effect: the high-frequency modes of the lattice form discrete sub-bands due to the finite lattice size [see Fig. 1d, e]. In computer simulations, we obtained a similar maximum value of $\chi \approx 2.4$ for the nontrivial lattice, whereas a comparable left-handed NLTL of the usual design (containing only identical nonlinear capacitances) has maximum $\chi \approx 0.47$ (see Supplementary Note 7).

The trivial lattice exhibits a much weaker third-harmonic signal. As indicated in Fig. 3c, for certain choices of $f_{\mathrm{in}}$ and $V_{\mathrm{in}}$, the value of $\chi$ in the nontrivial lattice is 200 times that in the trivial lattice. Figure 3d plots the normalized third-harmonic signal intensities versus the site index $k$, showing that they do not decay exponentially away from the edge. In the nontrivial lattice,

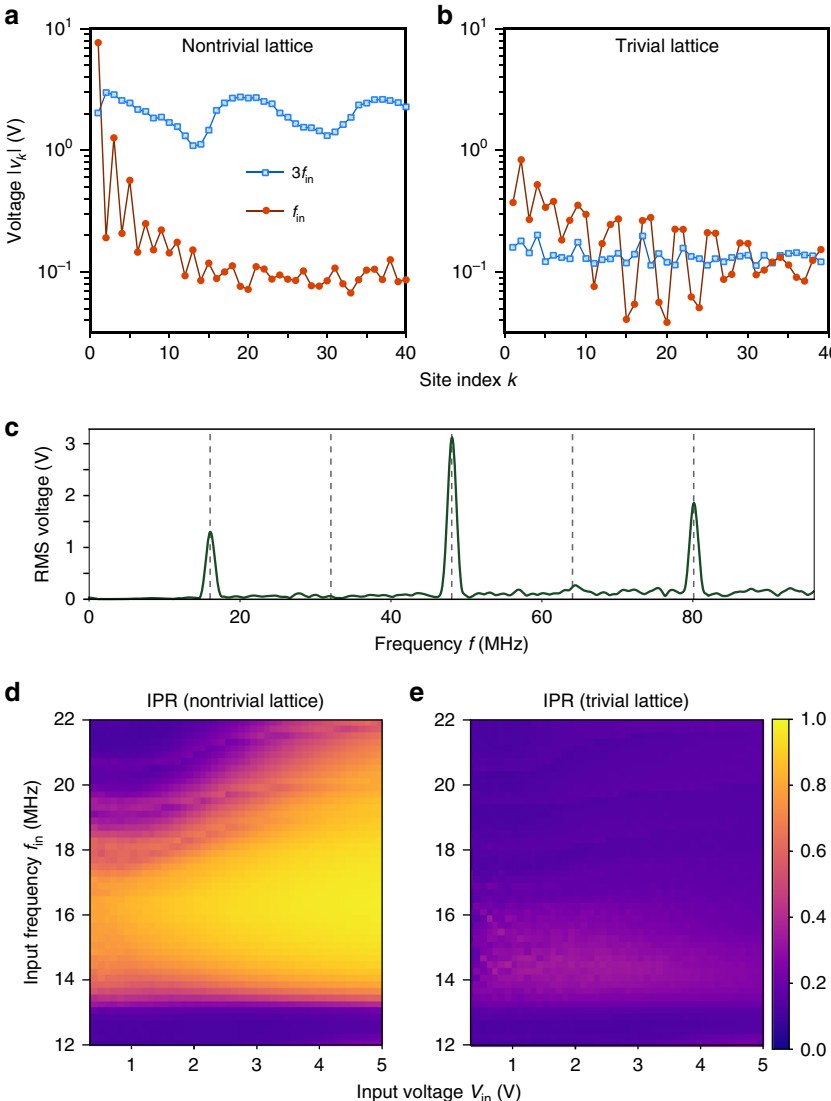

**Fig. 2** Experimental observation of harmonic generation in the nonlinear circuit. **a**, **b** Magnitude of the first- and third-harmonic voltage signals measured at different lattice sites, for **a** the nontrivial lattice, which has an SSH-like edge state in the linear limit, and **b** the trivial lattice, which has no edge state in the linear limit. The sinusoidal input signal, applied at the lattice edge (site 0), has frequency $f_{in} = 16$ MHz and amplitude $V_{in} = 2.5$ V. **c** Measured spectrum at site 3 for the nontrivial lattice corresponding to **a**. **d**, **e** Plot of the inverse participation ratio (IPR) versus input frequency $f_{in}$ and input voltage amplitude $V_{in}$, calculated from experimental measurements of the first-harmonic signal in the **d** nontrivial lattice and **e** trivial lattice. Here, $f_{in}$ is measured in steps of 0.2 MHz, and $V_{in}$ in steps of 0.1 V

the normalized third-harmonic signal increases with $V_{in}$ (i.e., stronger nonlinearity).

## Discussion

Our results point to a complex interplay between the topological edge state and higher-harmonic modes in the SSH-like NLTL. When a topological edge state exists in the linear lattice, it can be excited by an input signal at frequencies matching the bandgap of the linear lattice. The importance of the edge state is evident from the comparisons between the topologically trivial and nontrivial lattices (Figs. 2, 3). Note also that when the excitation frequency lies outside the linear bandgap, the two lattices behave similarly and the harmonic generation is relatively weak.

In the topologically nontrivial lattice, the resonant excitation generates third- and higher-harmonic signals that penetrate deep into the lattice, unlike the first-harmonic mode which is localized to the edge. Away from the edge, the higher-harmonic signals become stronger than the first harmonic, and hence dominate

the effective value of the nonlinear $\alpha$ parameter. In the linear lattice, $\alpha$ is the parameter that drives the topological transition, and increasing $\alpha$ leads to a larger bandgap and hence a more confined edge state. In the nonlinear regime, Fig. 2 shows an order-of-magnitude increase in the third-harmonic signal amplitude in the nontrivial lattice, relative to the trivial lattice; this implies an effective increase in $\alpha$, and indeed we see that the first-harmonic mode profile is more strongly localized. A more localized edge state, in turn, produces a stronger response to an input signal.

The above interpretation is supported by a more detailed analysis of the coupled equations governing the different circuit mode harmonics (see Supplementary Notes 2–4). These equations involve an effective $\alpha$ parameter whose approximate value, in the $n$-th unit cell, is $\langle \alpha_n \rangle \approx A + 2B \sum_m |W_n^m|^2$, where $|W_n^m|$ is the $m$-th harmonic of the bias voltage on the nonlinear capacitor in the $n$-th unit cell, and $m = 1, 3, 5, \ldots$ We are able to show that propagating waves can be self-consistently realized for higher

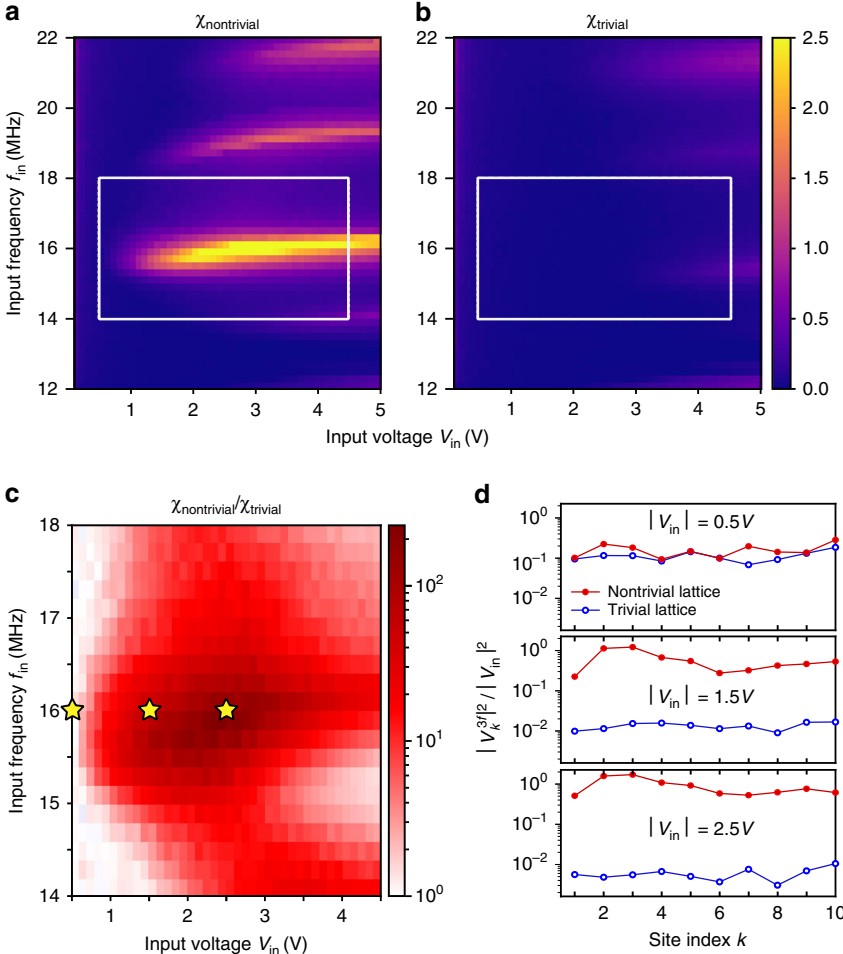

**Fig. 3** Measured third-harmonic signal intensities in the nonlinear circuit. **a**, **b** Normalized third-harmonic signal intensity $\chi$ versus input frequency $f_{in}$ and input voltage $V_{in}$, for the **a** nontrivial lattice and **b** trivial lattice. Here, $\chi$ is derived from experimental data using the definition (6). **c** Ratio of the trivial and nontrivial intensities, $\chi_{nontrivial}/\chi_{trivial}$, within the region indicated by boxes in **a** and **b**. **d** Measured third-harmonic intensities (normalized to the input signal) at different sites, for the three sets of input parameters indicated by stars in **c**: $V_{in} = 0.5$ V, 1.5 V, and 2.5 V, with fixed $f_{in} = 16$ MHz

($m \geq 3$) harmonics in the presence of non-linearity, even if the fundamental ($m = 1$) mode only has decaying solutions. The first-harmonic mode is localized to the edge, with localization length decreasing with $\langle \alpha_n \rangle$ in a manner similar to the linear SSH-like lattice. The generation of the higher-harmonic signals occurs mainly near the edge of the lattice, where the first-harmonic mode is largest. The nonlinearity-induced harmonic generation is aided by the well-known fact that the SSH edge state changes sign in each unit cell, corresponding to the fact that the gap closing in the SSH model takes place at the corner of the Brillouin zone[20]. This feature increases the bias voltages across the nonlinear capacitors, which can thus exceed the values of the voltages at individual sites.

The input signal can also be applied to the middle of the lattice. In this context, it is interesting to note that when we choose to excite a single site in the bulk of an SSH-like lattice, the sections to either side of the excitation have different topological phases: either trivial on the left and nontrivial on the right, or vice versa, depending on the two possible choices of excitation site. If the source impedance is sufficiently low, the effect is similar to exciting independent chains to the left and right; thus, the enhanced higher-harmonic signal is preferentially emitted toward the topologically nontrivial side (see Supplementary Note 6).

The presence of higher-harmonic signals distinguishes our system from previous studies of nonlinear topological edge states,

which were based on nonlinear self-modulation at a single harmonic. For instance, in a nonlinear SSH lattice where the coupling depends on the local intensity of a single mode, soliton-like edge states with anomalous mode profiles were predicted[21], and subsequently verified using a NLTL-like circuit[8]. That circuit, unlike ours, had narrow frequency bands and thus did not support propagating higher-harmonic modes. Topological solitons based on nonlinear self-modulation are also predicted to exist in higher-dimensional lattices[22–27]. In our case, the effective value of $\alpha$ away from the edge is dominated by the higher-harmonic signals; from the point of view of the first-harmonic mode, these act as a nonlocal nonlinearity, driving the entire lattice deeper into the topologically nontrivial regime, not just the sites with large first-harmonic intensity.

Our work opens the door to the application of topological edge states for enhancing harmonic generation, not just in transmission line circuits, but also a variety of other interesting systems. These include two-dimensional electronic lattices, where topological edge states have already been observed in the linear regime[2], and the unidirectional nature of the edge states may be even more beneficial for frequency-mixing[28]. Higher dimensional circuit lattices may possess different thresholds for bulk propagation in different directions, with an extreme generalization being that of a corner mode circuit constructed in ref. [4]. Electronic circuits incorporating amplifiers and resistances may also be able to

explore behaviors analogous to topological lasers[44–48], combining topological states with both nonlinearity and non-Hermiticity. Finally, circuits containing varactors that are explicitly time-modulated may be suitable for generating synthetic dimensions to realize topological features in higher dimensions[49–54].

## Methods

**Sample fabrication and experimental procedure**. The NLTL was implemented on a PCB (Seeed Tech. Co.), with each nonlinear capacitor consisting of a pair of back-to-back varactors (Skyworks Solutions, SMV1253-004LF). The transmission line, as fabricated, is topologically nontrivial, as shown in Fig. 1a. To probe the trivial circuit, we use a switch to add one sublattice unit cell at the rightmost end of the transmission line, and disconnect the leftmost $C_a$ and $L$ in Fig. 1a. This yields a nontrivial circuit of same length, with the $C_a$ and $C_b$ capacitors swapped.

A function generator (Tektronix AFG3102C) supplies the continuous-wave sinusoidal input voltage, and the voltages on successive lattice sites, $k \geq 1$, are measured by an oscilloscope (Rohde & Schwarz RTE1024) in high-impedance mode. Numerical results were obtained using the SPICE circuit simulator.

## Data availability

Raw experimental data and Python code used to generate all plots can be found at https://doi.org/10.21979/N9/I74ZP1. All other data are available from the authors upon reasonable request.

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

## Acknowledgements

The authors are grateful to H. Wang, D. Leykam, and Z. Gao for helpful discussions. Y.W., L.-J.L., B.Z., and Y.D.C. were supported by the Singapore MOE Academic Research Fund Tier 2 Grant MOE2015-T2-2-008, the Singapore MOE Academic Research Fund Tier 3 Grant MOE2016-T3-1-006, and the Singapore MOE Academic Research Fund MOE2018-T2-1-022(S).

## Author contributions

Y.W. designed the printed circuit board and performed the measurements. Y.W. and L.-J.L. analyzed the data. B.Z. and Y.D.C. supervised the project. Y.W., L.-J.L., C.H.L., B.Z., and Y.D.C. participated in the design and interpretation of the experiment, and all contributed substantially to the work.

## Additional information

**Competing interests:** The authors declare no competing interests.

