## [Peer Review File · Nature Communications]

Reviewers' comments:

Reviewer #1 (Remarks to the Author):

In this manuscript, the authors detail a realization of a SSH-model using an electronic transmission line with time-dependent varactor modulators. The model is known to have two topologically distinct phases where additional boundary modes appear in the nontrivial case. The circuit design details and relevant approximations are described in the main text and are supported by supplemental material. The details of the experiment are then presented, where the presence of the in-gap topological boundary mode allows for an excitation of the system such that sizable third harmonic generation is produced. This is a very nice realization that brings topology in classical systems towards the contemporary realm of studying the impact on nonlinearities on the system. I would, however, not recommend publication until the authors clarify the following points:

1. There are several works in photonic systems that discuss lasing from the boundary of an SSH chain [Phys. Rev. B 93, 195317 (2016), Science 359, eaar4003 (2018), Science 359, eaar4005 (2018), Science 358, 636 (2017), Nature Photonics 11, 651 (2017)], where higher harmonic generation should also occur. Considering that one of the authors of this work is an author of one of the works that I allude to, it would be nice to have a clearer relationship between the electric and photonic systems that realize very similar physics.
2. The third harmonic is effectively generated by the parametric time-dependence of the varactors. How does the current result tie up with the idea of synthetic dimensions, see, e.g., [Phys. Rev. Lett. 108, 133001 (2012), Phys. Rev. Lett. 112, 043001 (2014), Science 349, 1510–1513 (2015), Science 349, 1514–1518 (2015), Phys. Rev. A 93, 043827 (2016), Phys. Rev. Lett. 115, 195303 (2015), arxiv:1807.11468].
3. The formation of a propagating soliton in an SSH model was also studied for mechanical systems, how does this work relate to this work [Nature Physics 10, 39 (2014)], as well as to follow up results.
4. Last, the SSH model exhibits protected boundary modes that are pinned to appear in the middle of the spectral gap (and at zero energy). In Fig. 1(d), this is not the case. How can the authors explain this?

Reviewer #2 (Remarks to the Author):

The authors have demonstrated topologically enhanced third harmonic generation from a nonlinear transmission line with topologically edge state. The work is very interesting and potentially important as it shows how topology can be applied in applications, especially in nonlinear optics. In my opinion, this work, with the generic principle demonstrated, is attractive to a broad readership and will generate a series of future works as well. I believe this work can be considered for publication after the authors have addressed the following:

Can you plot the band structure ω against k for two particular α s that one in trivial and one in non-trivial region? That will help to see whether we have negative index. Are there any importance for the backward waves supported by the negative index medium?

You are exciting the lattice (with $\alpha > 1$) in trivial and non-trivial state at one end depending on whether you excite at site A or B. It's interesting that you do not need to fabricate different samples to choose the two configurations. Can you elaborate more on this point? Suppose I choose the excitation in the middle, like the 4th unit cell, what will happen?

Is the IPR enhancement really related to the edge state or not? At $\alpha = 1.3$, the gap is from

around 13 to 16 MHz while the edge state is only at a single frequency (around 14.5 MHz, reading from fig. 1d) but the IPR seems to have a broadband behavior from 13.5 to 18 (stepping into the bands). The large bandwidth seems indicates that the IPR enhancement is not just corresponding to edge state (for larger excitation of edge state to give stronger localization), but if it is about the non-trivial bandgap itself, what's the mechanism?

Similarly, can you clarify whether χ_{\max} at 16Mhz in fig. 3a corresponds to edge state?

Is χ being larger than one means the power in the 3rd harmonic signal is even bigger than the power of the fundamental frequency?

Response to Reviewer Comments for ‘Topologically Enhanced Harmonic Generation in a Non-linear Transmission Line Metamaterial’

We are grateful to the Reviewers for their careful reading of the manuscript. Below, we provide in-depth replies to specific points raised by the Reviewers.

Reviewer 1

Q1: There are several works in photonic systems that discuss lasing from the boundary of an SSH chain [Phys. Rev. B 93, 195317 (2016), Science 359, eaar4003 (2018), Science 359, eaar4005 (2018), Science 358, 636 (2017), Nature Photonics 11, 651 (2017)], where higher harmonic generation should also occur... it would be nice to have a clearer relationship between the electric and photonic systems that realize very similar physics.

In the revised manuscript’s Discussion section, we have added more discussion about how our work relates to SSH lasers (the paragraph beginning ‘*Our work opens the door to the application of topological edge states for enhancing harmonic generation, not just in transmission line circuits, but also a variety of other interesting systems...*’). The papers mentioned by the Reviewer are now cited.

As the Reviewer notes, SSH lasers are of substantial recent interest, and lasers are inherently nonlinear due to the role of the gain medium. There are, however, significant differences. Lasers are non-Hermitian, and the combination of nonlinearity and gain leads to the spontaneous emergence of laser modes with specific frequencies. In our system, the harmonics are generated from a tunable-frequency input signal. Certainly, it is interesting to look into topologically-aided harmonic generation in nonlinear optical (e.g., χ_3) media, either with the SSH model or other structures (e.g. 2D lattices). Our explorations in electronics (where fabrication and analysis is easier) may serve as a useful guide for that effort.

One important aspect highlighted by the present work (as well as previous studies) is that the transport properties of the fundamental harmonic and higher harmonics have to be carefully planned for. Our left-handed transmission line supports higher harmonic modes up to high frequencies, whereas right-handed transmission lines would not provide circuit modes for the higher harmonics to couple into. In photonic devices, higher harmonic signals could couple to the continuum (free space modes), instead of being tightly bound to the lattice or cavity [e.g. Science 359, eaar4005 (2018)], a possibility that does not exist for electronic circuits of the present type.

Q2: The third harmonic is effectively generated by the parametric time-dependence of the varactors. How does the current result tie up with the idea of synthetic dimensions, see, e.g., [Phys. Rev. Lett. 108, 133001 (2012), Phys. Rev. Lett. 112, 043001 (2014), Sci-

ence 349, 1510–1513 (2015), *Science* 349, 1514–1518 (2015), *Phys. Rev. A* 93, 043827 (2016), *Phys. Rev. Lett.* 115, 195303 (2015), *arxiv:1807.11468*].

This is an interesting point, and we thank the Reviewer for raising it. Schemes that use nonlinearity to generate synthetic dimensions require specially designing both the nonlinearity and lattice. The couplings between different harmonics play the role of hoppings between sites, and this occurs through parametric time dependences related to (but not quite the same as) the parametric amplification processes in our experiment. In order to discretize the synthetic dimension, the resonators should have discrete, spaced-out resonances, which is not the case for our transmission line model.

Our present circuit thus lacks the necessary features for interpretation in terms of synthetic dimensions. However, it is possible that a similar circuit could realize synthetic dimensions (e.g., by applying explicit time modulations to the varactor biases to produce the desired inter-frequency couplings). We added a note to this effect, with the references suggested by the Reviewer, in the Discussion section.

Q3: The formation of a propagating soliton in an SSH model was also studied for mechanical systems, how does this work relate to this work [Nature Physics 10, 39 (2014)], as well as to follow up results.

That paper describes how mechanical lattices can contain zero modes that satisfy an index theorem, and are robust to local perturbations. These zero modes can be mapped to the topological boundary modes of the *linear* SSH model (Eq. 2 of that paper maps to the linear SSH Hamiltonian). Although the full mechanical model is nonlinear, it was linearized for the analysis. The fully nonlinear mechanical model would support Jackiw-Rebbi-like soliton modes, but that was not analyzed in that paper, and solitons are in any case not the major emphasis of our present work (which focuses on harmonic generation). We have added this paper to our list of cited classical analogues of topological insulators.

Q4: the SSH model exhibits protected boundary modes that are pinned to appear in the middle of the spectral gap (and at zero energy). In Fig. 1(d), this is not the case. How can the authors explain this?

This is because circuit equations don't *directly* form the SSH eigenvalue problem. Instead, they can be mapped one-to-one to the eigenvalue problem. In the figure in the manuscript, we are using the physical frequency variable, ω , as the vertical axis. But we can also convert the frequency into a quasi-energy

$$E = 1 + \frac{1}{\alpha} - \frac{\omega_a^2}{\omega^2}$$

as indicated in Eq. (2). The resulting quasi-energy spectrum is shown below. The boundary modes are indeed in the middle of the gap, and pinned to $E = 0$. We added a brief note to the revised manuscript, explaining this explicitly, below Eq. (4).

Figure 1: Quasi-energy spectrum for the nontrivial and trivial SSH circuits

Reviewer 2

Q1: Can you plot the band structure ω against k for two particular α s that one in trivial and one in non-trivial region? That will help to see whether we have negative index. Are there any importance for the backward waves supported by the negative index medium?

We have revised Fig. S1 (in the Supplemental Material) to include a plot of the bulk bandstructure. The upper band (which extends to $\omega \rightarrow +\infty$) indeed has negative dispersion, for both topological phases ($\alpha > 1$ and $\alpha < 1$).

The advantages provided by the negative dispersion are as follows. First, in left-handed transmission lines, the negative dispersion is associated with having an unbounded band that extends to $\omega \rightarrow +\infty$, and this is helpful to allow the higher harmonics generated by the first harmonic (at the bandgap frequency) to coincide with circuit modes. Second, harmonic generation in standard right-handed transmission lines is known to be susceptible to shock wave formation, which spoils the ‘clean’ parametric interactions we are interested in, and the negative dispersion suppresses shock waves [J. Phys. D: Appl. Phys. 41, 173001 (2008)].

Q2: You are exciting the lattice (with $\alpha > 1$) in trivial and non-trivial state at one end depending on whether you excite at site A or B. It’s interesting that you do not need to fabricate different samples to choose the two configurations. Can you elaborate more on this point? Suppose I choose the excitation in the middle, like the 4th unit cell, what will happen?

To probe the trivial circuit, we use a switch to add one sublattice unit cell at the tail (rightmost end) of the transmission line, and remove the first sublattice unit cell (leftmost C_a and L). We then apply the source at the point B labelled in Fig. 1a of the manuscript. This yields a nontrivial circuit (of same length), with the C_a and C_b capacitors swapped.

To convey this important point more clearly, we have added some new text to the Methods section of the revised manuscript. We thank the Reviewer for raising the issue.

As for the question about exciting the circuit in the middle: if the voltage source has sufficiently low impedance, the left and right parts of the lattice act as two independent, shorter, transmission lines. This may be interesting, as one section will be trivial and the other nontrivial, which means that the higher harmonic signal will propagate mainly to the left or to the right, depending on the choice of excitation site. This follows directly from the behavior of the edge-excitation case, but we are grateful to the Reviewer for helping to tease it out; we have added a paragraph to the Discussion section describing this (*‘If the input signal is applied to the middle of the lattice. . .’*). We have also added a new Section S2E to the Supplemental Material, which provides more details and briefly mentions the (more complicated) behavior for high input impedance.

Q3: Is the IPR enhancement really related to the edge state or not? At $\alpha = 1.3$, the gap is from around 13 to 16 MHz while the edge state is only at a single frequency (around 14.5 MHz, reading from fig. 1d) but the IPR seems to have a broadband behavior from 13.5 to 18 (stepping into the bands). The large bandwidth seems indicates that the IPR enhancement is not just corresponding to edge state (for larger excitation of edge state to give stronger localization), but if it is about the non-trivial bandgap itself, what's the mechanism?

The IPR enhancement is caused by the effect of the strong nonlinearly generated third harmonic signal on the band gap at the first harmonic. First, note that in the linear lattice, the decay profile is the same with or without the edge state, and only depends on the width of the bandgap. The presence of the edge state then causes a resonant increase in the amplitude of the voltage amplitude signal (without altering the profile and thus the IPR). As shown by the characteristic curve in Fig.~1(c), the nonlinear capacitance drops rapidly with increasing voltage bias (even as low as 0.1 V), resulting in a larger effective value of α .

This process is explained in the discussion accompanying Fig. 2 of the manuscript (the paragraph above Eq. (6)), and also shown explicitly by Fig. S2(c) in the Supplemental Material.

Figure 2: Voltage amplitude on the left lattice edge (left) and IPR (right) versus driving frequency, for a linear circuit with $\alpha = 3$ driven by a fixed-amplitude (1 V) voltage source.

To emphasize this point, the above figure plots both the voltage amplitude on the left edge and the IPR, versus driving frequency, for a linear circuit with $\alpha = 3$ (i.e., in the topologically nontrivial phase). This circuit is derived by replacing our nonlinear capacitors with linear $15.7 \text{ pF} = 47 \text{ pF}/\alpha$ capacitors, and all circuit resistances removed. The plotted frequency range corresponds to the circuit bandgap. We find that the voltage amplitude peaks around 16 MHz, over a frequency range much narrower than the bandgap. The IPR, however, remains large over the entire frequency range corresponding to the bandgap. (The inclusion of circuit resistances smooths out these curves slightly, but does

not produce any qualitative difference.)

In the nonlinear circuit, the presence of the topological edge state gives rise to a strong higher-harmonic signal, which propagates into the lattice. This, in turn, enhances the effective bandgap seen by the first harmonic mode, making it more localized. This is discussed in the first paragraph of the Discussion section. In the revised manuscript, we have tweaked the explanation to make it clearer.

Q4: Similarly, can you clarify whether χ_{max} at 16Mhz in fig. 3a corresponds to edge state? Is χ being larger than one means the power in the 3rd harmonic signal is even bigger than the power of the fundamental frequency?

We believe that the relatively narrow range in the χ plot is not due specifically to the edge state, but instead is a finite-size effect. As with any finite lattice, the sub-bands that make up the upper band become sparser at high frequencies (e.g., for 16 MHz driving signal, the third harmonic frequency is 48 MHz). This can be observed in Fig. 1(d) and (e).

The presence of the topological edge state (near 16MHz with high input voltage or large effective α) allows for a strong higher-harmonic signal to be produced, but the higher-harmonic signals still need to coincide with a sub-band. Hence, we only observe enhanced χ in a relative small frequency range. This is explained in the manuscript in the discussion accompanying Fig. 3: ‘*The fact that χ peaks over a relatively narrow frequency range... may be a finite-size effect...*’

The argument is further supported by the SPICE simulation results shown in the figure below. In this set of simulations, we first sweep the source frequency over $f_1 = 12 \rightarrow 22$ MHz, and the voltage amplitude over $V_1 = 0 \rightarrow 5$ V. Next, we repeated the simulations with an additional weak signal at the third harmonic frequency $f_3 = 3f_1$, with voltage amplitude $V_3 = V_1/100$. The additional perturbation at f_3 is too weak to influence the circuit nonlinearity significantly, but it ‘sees’ the nonlinear effects caused by the main signal at frequency f_1 . The heat map plots the difference in χ (the figure of merit for the signal intensity at f_3 , defined in the same way as in the paper) between the two cases. If there were no finite size effect, the additional signal at f_3 should couple more-or-less equally well over the plotted frequency range, so we would expect a very broad peak. Instead, we observe a narrowly-peaked signal, indicating that the additional signal can only couple to the circuit at specific frequencies; the frequency widths are also qualitatively similar to the χ plot shown in the main text.

Having $\chi > 1$ does not necessarily mean the third harmonic power is bigger than the fundamental frequency. In the definition of χ , the averaged third harmonic signal is normalized to the input voltage of the source; it is a relative value representing how efficiently the input signal is being converted to the third harmonic signal. However, within the lattice, the third harmonic signal intensity *does* become larger than the first harmonic signal, in the topologically nontrivial case. This is shown in Fig. 2(a) of the manuscript.

Figure 3: Colormap plot showing the difference in the normalized third harmonic intensity χ , with and without an additional weak signal added at the third harmonic frequency.

REVIEWERS' COMMENTS:

Reviewer #1 (Remarks to the Author):

I thank the Authors for their elaborate reply. Following their revision, I can now recommend publication in Nature Communication.

Reviewer #2 (Remarks to the Author):

The authors have addressed my concerns. In particular, they have clarified the advantage of using negative indices in getting the high harmonics to fall into circuit modes. Frequency dispersions are also added.

They have also clarified how the circuits are excited selectively, even with new simulation results added for the excitation at different sites. It surely enriches the physics and applications. The relationship between the edge state and a broadband enhancement of the signal is also addressed.

I would like to support its publication. The current manuscript demonstrates how topology generates applications in nonlinear dynamics.